META-RESEARCH

# Blinding reduces institutional prestige bias during initial review of applications for a young investigator award

**Abstract** Organizations that fund research are keen to ensure that their grant selection processes are fair and equitable for all applicants. In 2020, the Arnold and Mabel Beckman Foundation introduced blinding to the first stage of the process used to review applications for Beckman Young Investigator (BYI) awards: applicants were instructed to blind the technical proposal in their initial Letter of Intent by omitting their name, gender, gender-identifying pronouns, and institutional information. Here we examine the impact of this change by comparing the data on gender and institutional prestige of the applicants in the first four years of the new policy (BYI award years 2021–2024) with data on the last four years of the old policy (2017–2020). We find that under the new policy, the distribution of applicants invited to submit a full application shifted from those affiliated with institutions regarded as more prestigious to those outside of this group, and that this trend continued through to the final program awards. We did not find evidence of a shift in the distribution of applicants with respect to gender.

**ANNE E HULTGREN\*, NICOLE MF PATRAS, JENNA HICKS**

## Introduction

Studies on the impact of blinding in peer review, including studies that either fully remove the identities of applicants or use other methods to mask or change perceived applicant identity, have shown mixed results as to the benefits of blinding with respect to bias against certain populations. With respect to gender, the range in outcomes include studies that show a reduction in bias with blinded reviews (*Johnson and Kirk, 2020*; *Goldin and Rouse, 2000*), those that show no effect between blinded or unblinded reviews (*Tomkins et al., 2017*; *Forscher et al., 2019*; *Marsh et al., 2008*; *Ross et al., 2006*), and those that find unblinded reviews have less bias in their outcomes (*Ersoy and Pate, 2022*). Studies evaluating outcomes with respect to race similarly show a range of outcomes from reduction in bias with blinded reviews (*Nakamura et al., 2021*), to no effect between blinded or unblinded reviews (*Forscher et al., 2019*).

Other studies have examined if blinding can reduce bias with respect to author institutional affiliation, and several studies have shown reduction in bias towards highly prestigious institutions and authors (*Tomkins et al., 2017*; *Ross et al., 2006*; *Ersoy and Pate, 2022*; *Nakamura et al., 2021*; *Sun et al., 2022*). Additionally, one study that examined a process in which an initial blinded review was followed with an unblinded review, found that the reviewers were more likely to increase their scores in the second (unblinded) stage if the author was considered to be from an institution with additional resources and have an author with extensive prior experience, as evidenced through the comments collected from the reviewers (*Solans-Domènech et al., 2017*). Notably, these studies have used different proxies for 'institutional prestige', including published ranked lists of institutions from independent entities (*Tomkins et al., 2017*; *Ross et al., 2006*), age of the institution itself (*Marsh et al., 2008*), list of institutions chosen by the study authors (*Ersoy and Pate, 2022*), publication records (*Sun et al., 2022*), and size of the institution by student population (*Murray et al., 2016*).

\*For correspondence:
ahultgren@beckman-foundation.org

**Competing interest:** The authors declare that no competing interests exist.

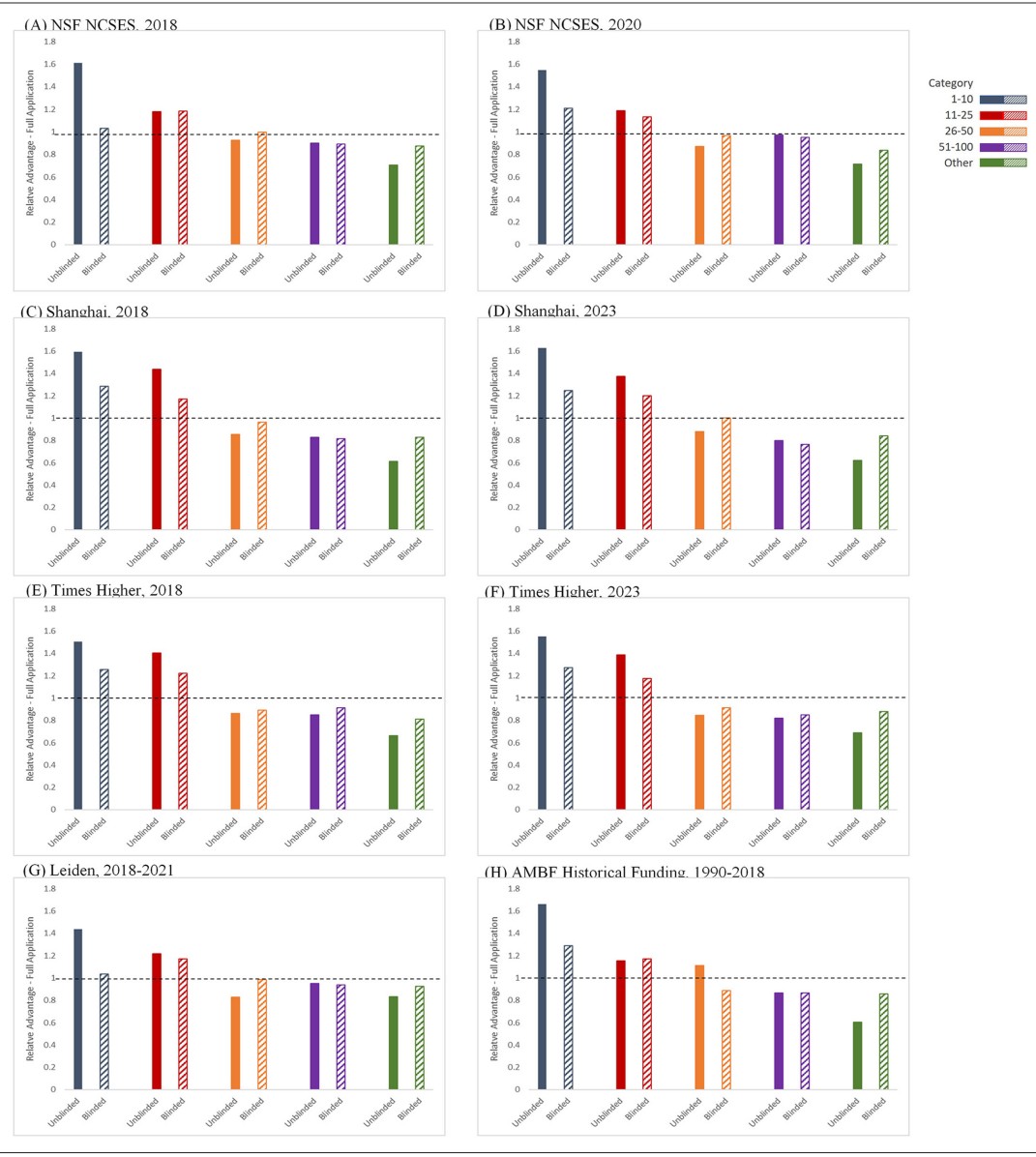

**Figure 1.** Relative Advantage–Full Application. Ratio of the percentage of LOI applicants in different institutional categories receiving an invitation to submit a Full Application, compared to the percentage of any LOI applicant receiving an invitation to submit a Full Application during unblinded reviews (solid bars, left) and blinded reviews (hatched bars, right). The eight different institutional rankings used in the study were: (**A**) NCSES/NSF-2018; (**B**) NCSES/NSF-2020; (**C**) Shanghai Ranking-2018; (**D**) Shanghai Ranking-2023; (**E**) Times Higher-2018; (**F**) Times Higher-2023; (**G**) CWTS Leiden: 2018–2021; (**H**) AMBF historical funding: 1990–2018.

The online version of this article includes the following source data and figure supplement(s) for figure 1:

**Source data 1.** BYI LOIs, Full Application Invitations, and Program Awards by institutional category.

**Figure supplement 1.** Relative Advantage–Award.

The mission of the Arnold and Mabel Beckman Foundation (AMBF) is to fund innovative research projects, especially those that open new avenues of research in chemistry and life sciences. In particular, the foundation's Beckman Young Investigators (BYI) program seeks to satisfy Dr Arnold O Beckman's directive to 'support young scientists that do not yet have the clout to receive major federal research grants,' and we strive to support outstanding young scientists who are moving into new, transformative areas of scientific inquiry. It stands to reason that our mission could be more effectively fulfilled if we ensure that our review process is insulated as much as

**Table 1.** Numbers of Letters of Intent (LOIs), Full Application Invitations, and BYI Program Awards for 2017–2024.

| | Unblinded (2017–2020) | | | | Blinded (2021–2024) | | | |
|---|---|---|---|---|---|---|---|---|
| Year | 2017 | 2018 | 2019 | 2020 | 2021 | 2022 | 2023 | 2024 |
| LOIs Reviewed | 316 | 351 | 405 | 293 | 256 | 230 | 194 | 246 |
| Full Application Invitations | 99 | 100 | 108 | 98 | 96 | 105 | 94 | 97 |
| Program Awards | 8 | 10 | 10 | 10 | 11 | 10 | 11 | -- |

possible from implicit or explicit gender or institutional prestige bias by reviewers.

In this article we report the results of a multi-year study to assess the impact of blinding gender and institutional affiliation in the first stage of the application process for BYI awards, which involves applicants submitting a technical proposal as part of an initial Letter of Intent (LOI). In 2020 applicants were instructed to blind their technical proposals by omitting their name, gender, gender-identifying pronouns, and institutional information. We have examined the impact of this change in policy by comparing data on all stages of process (from the initial review through to the award of grants) for the four years before the new policy was introduced (2017–2020) and the first four years of the new policy (2021-2024). We were not able to perform a similar analysis with regards to race as our application process during unblinded review years did not collect race or ethnicity information from applicants.

Throughout the years included in this study, 2017–2024, the criteria for applicant eligibility as well as the overall review process were not substantively changed. Thus, in the absence of institutional prestige or gender bias in our review process, the distribution of advancing LOIs and program awards before and after blinding should be similar. This was the case when considering gender distribution, providing no evidence of gender bias in either the unblinded or the blinded reviews. However, upon blinding at the LOI stage, we did find a reduction in the relative advantage of more prestigious institutions in advancing to a full application invitation and in receiving a program award. We therefore conclude that there was an institutional prestige bias in our review process, and that blinding helped to reduce the impact of that bias. This reduction in bias brings us closer to our goal of the equitable allocation of research funding resources based on scientific merit that is not influenced by implicit or explicit institutional prestige bias from reviewers.

*Study limitations*

The application and review processes described in this study were conducted during the normal operations of AMBF, which introduced several limitations to the study. We neither requested from applicants, nor created ourselves, blinded and unblinded versions of the same applications to review the same set of research proposals in both formats to compare outcomes. We also did not explicitly test if the institution had a direct impact on reviews by, for example, taking a research proposal from a highly ranked institution and changing the affiliation of the applicant to a lower ranked institution, and vice-versa, in order to compare review outcomes with the original and modified institutional affiliations.

In addition, we appoint a new set of reviewers to assist with the BYI review process every year, and therefore the mix of reviewer technical expertise and potential biases varies from year to year, and we do not have multi-year data using the same set of reviewers. We also did not request reviewers to provide a list of institutions that they perceive to have high prestige and therefore the ranked lists we used in this study may or may not accurately reflect the implicit biases of our reviewers.

Finally, our unblinded applications did not request applicants to self-report their gender and so the data used in the unblinded gender analysis was curated manually from the applicant's name which may have led to some errors in gender assignments. In 2020, AMBF began collecting self-reported gender from applicants for our internal use and those data were used for the later years of this study.

## Results

The total number of reviewed LOIs, Full Applications invitations, and BYI Program Awards in program years 2017–2020 (unblinded) and 2021–2024 (blinded) is presented in *Table 1*; the 2024

**Table 2.** Relative Advantage–Full Application.

The average value and ranges by institutional category, with Chi-squared association test and Cramer's V statistic of unblinded and blinded LOI reviews for full application invitations, for the eight institutional rankings used in the study.

**Ranked List: NCSES 2018**

| Category | Unblinded Average (Range) | Blinded Average (Range) | Unblinded - Blinded |
|---|---|---|---|
| 1–10 | 1.6 (1.4–1.9) | 1.0 (1.0–1.1) | 0.6 |
| 11–25 | 1.2 (1.0–1.3) | 1.2 (1.0–1.4) | 0 |
| 26–50 | 0.93 (0.70–1.3) | 1.0 (0.86–1.1) | –0.07 |
| 51–100 | 0.90 (0.66–1.1) | 0.89 (0.79–0.95) | 0.01 |
| Other | 0.70 (0.60–0.82) | 0.87 (0.69–1.1) | –0.17 |
| Analysis | Unblinded | Blinded | Unblinded - Blinded |
| Chi-squared | 48.64 | 9.47 | 39.17 |
| p (d.f.=4) | 6.95E-10 | 0.0503 | |
| Cramer's V | 0.19 | 0.1 | 0.09 |
| Effect Size | Medium | Small | |

**Ranked List: NCSES 2020**

| Category | Unblinded Average (Range) | Blinded Average (Range) | Unblinded - Blinded |
|---|---|---|---|
| 1–10 | 1.5 (1.3–1.7) | 1.2 (1.1–1.3) | 0.3 |
| 11–25 | 1.2 (1.1–1.2) | 1.1 (0.9–1.3) | 0.1 |
| 26–50 | 0.87 (0.63–1.3) | 0.97 (0.86–1.1) | –0.1 |
| 51–100 | 0.97 (0.65–1.2) | 0.95 (0.89–1.0) | 0.02 |
| Other | 0.71 (0.61–0.82) | 0.84 (0.65–0.99) | –0.13 |
| Analysis | Unblinded | Blinded | Unblinded - Blinded |
| Chi-squared | 44.9 | 10.35 | 34.55 |
| p (d.f.=4) | 4.17E-09 | 0.0349 | |
| Cramer's V | 0.18 | 0.11 | 0.07 |
| Effect Size | Medium | Small | |

**Ranked List: Shanghai Ranking 2018**

| Category | Unblinded Average (Range) | Blinded Average (Range) | Unblinded - Blinded |
|---|---|---|---|
| 1–10 | 1.6 (1.5–1.8) | 1.3 (1.0–1.7) | 0.3 |
| 11–25 | 1.4 (1.4–1.7) | 1.2 (1.0–1.3) | 0.2 |
| 26–50 | 0.85 (0.80–0.91) | 0.96 (0.82–1.1) | –0.11 |
| 51–100 | 0.83 (0.64–1.0) | 0.82 (0.69–0.91) | 0.01 |
| Other | 0.61 (0.38–0.80) | 0.83 (0.61–1.1) | –0.22 |
| Analysis | Unblinded | Blinded | Unblinded - Blinded |
| Chi-squared | 71.87 | 17.84 | 54.03 |
| p (d.f.=4) | 9.14E-15 | 0.00133 | |
| Cramer's V | 0.23 | 0.14 | 0.09 |
| Effect Size | Medium | Small | |

**Ranked List: Shanghai Ranking 2023**

| Category | Unblinded Average (Range) | Blinded Average (Range) | Unblinded - Blinded |
|---|---|---|---|
| 1–10 | 1.6 (1.4–1.9) | 1.2 (1.1–1.4) | 0.4 |
| 11–25 | 1.4 (1.1–1.6) | 1.2 (1.1–1.3) | 0.2 |
| 26–50 | 0.88 (0.75–1.0) | 1.0 (0.92–1.1) | –0.12 |
| 51–100 | 0.80 (0.67–0.91) | 0.80 (0.62–0.86) | 0 |
| Other | 0.62 (0.43–0.76) | 0.84 (0.61–1.2) | –0.22 |
| Analysis | Unblinded | Blinded | Unblinded - Blinded |
| Chi-squared | 70.85 | 21.94 | 48.91 |
| p (d.f.=4) | 1.50E-14 | 0.000206 | |
| Cramer's V | 0.23 | 0.15 | 0.08 |
| Effect Size | Medium | Medium | |

**Ranked List: Times Higher 2018**

| Category | Unblinded Average (Range) | Blinded Average (Range) | Unblinded - Blinded |
|---|---|---|---|
| 1–10 | 1.5 (1.3–1.6) | 1.3 (1.2–1.4) | 0.2 |
| 11–25 | 1.4 (1.2–1.7) | 1.2 (1.1–1.3) | 0.2 |
| 26–50 | 0.86 (0.76–1.1) | 0.89 (0.83–0.96) | –0.03 |
| 51–100 | 0.85 (0.67–1.1) | 0.91 (0.73–1.1) | –0.06 |
| Other | 0.66 (0.55–0.76) | 0.81 (0.53–1.2) | –0.15 |
| Analysis | Unblinded | Blinded | Unblinded - Blinded |
| Chi-squared | 60.03 | 19.5 | 40.53 |
| p (d.f.=4) | 2.86E-12 | 0.000626 | |
| Cramer's V | 0.21 | 0.15 | 0.06 |
| Effect Size | Medium | Medium | |

**Ranked List: Times Higher 2023**

| Category | Unblinded Average (Range) | Blinded Average (Range) | Unblinded - Blinded |
|---|---|---|---|
| 1–10 | 1.6 (1.4–1.6) | 1.3 (1.2–1.4) | 0.3 |
| 11–25 | 1.4 (1.2–1.6) | 1.2 (1.2–1.2) | 0.2 |
| 26–50 | 0.84 (0.63–0.98) | 0.91 (0.77–1.1) | –0.07 |
| 51–100 | 0.82 (0.56–1.1) | 0.83 (0.81–0.92) | –0.01 |
| Other | 0.69 (0.62–0.75) | 0.88 (0.65–1.1) | –0.19 |
| Analysis | Unblinded | Blinded | Unblinded - Blinded |
| Chi-squared | 60.78 | 17.64 | 43.14 |
| p (d.f.=4) | 1.98E-12 | 0.00145 | |
| Cramer's V | 0.21 | 0.14 | 0.07 |
| Effect Size | Medium | Small | |

*Table 2 continued on next page*

*Table 2 continued*

| Ranked List: Leiden 2018–2021 | | | | Ranked List: AMBF 1990–2018 | | | |
|---|---|---|---|---|---|---|---|
| Category | Unblinded Average (Range) | Blinded Average (Range) | Unblinded - Blinded | Category | Unblinded Average (Range) | Blinded Average (Range) | Unblinded - Blinded |
| 1–10 | 1.4 (1.1–1.8) | 1.0 (0.88–1.2) | 0.4 | 1–10 | 1.7 (1.5–2.0) | 1.3 (1.2–1.3) | 0.4 |
| 11–25 | 1.2 (1.1–1.3) | 1.2 (1.1–1.2) | 0 | 11–25 | 1.2 (0.83–1.4) | 1.2 (1.1–1.2) | 0 |
| 26–50 | 0.83 (0.70–0.92) | 0.98 (0.89–1.1) | –0.15 | 26–50 | 1.1 (1.0–1.2) | 0.90 (0.70–1.1) | 0.2 |
| 51–100 | 0.95 (0.76–1.1) | 0.94 (0.78–1.3) | 0.01 | 51–100 | 0.86 (0.56–1.0) | 0.87 (0.70–1.0) | –0.01 |
| Other | 0.83 (0.65–1.0) | 0.93 (0.71–1.1) | –0.1 | Other | 0.60 (0.54–0.72) | 0.86 (0.70–1.2) | –0.26 |
| Analysis | Unblinded | Blinded | Unblinded - Blinded | Analysis | Unblinded | Blinded | Unblinded - Blinded |
| Chi-squared | 27.96 | 5.23 | 22.73 | Chi-squared | 74.22 | 20.34 | 53.88 |
| p (d.f.=4) | 1.27E-05 | 0.265 | | p (d.f.=4) | 2.91E-15 | 0.000427 | |
| Cramer's V | 0.14 | 0.08 | 0.06 | Cramer's V | 0.23 | 0.15 | 0.08 |
| Effect Size | Small | Small | | Effect Size | Medium | Medium | |

Program Awards have not been completed as of manuscript preparation.

### *Institutional prestige*

To examine the extent to which institutional prestige bias may have influenced our LOI review process, we developed eight institutional ranking schema, further divided into five institutional categories, based on published rankings of institutions from four independent organizations as well as from an internal analysis of historical AMBF funding trends, described further in the Methods section. We then calculated the Relative Advantage–Full Application across institutional categories for each schema: a ratio of the percentage of LOIs submitted in a particular institutional category that received an invitation to submit a full application to the percentage of all LOIs that received a full application invitation. If there was no implicit or explicit institutional prestige bias from our reviewers in the LOI review process, then we would expect the Relative Advantage–Full Application would be the same in the unblinded and blinded reviews. Therefore, we examined the difference in Relative Advantage–Full Application between the unblinded and blinded reviews to determine if there was institutional prestige bias in our review process. We repeated the analysis to determine the Relative Advantage–Award for each institutional category as a ratio of the percentage of LOIs submitted in a particular institutional category that received a program award to the percentage of all LOIs that received a program award. Again, we examined the difference in Relative Advantage–Award from unblinded and blinded reviews to determine if there was institutional prestige bias in our review process.

*Figure 1* shows the average Relative Advantage–Full Application afforded to an LOI applicant to receive an invitation to submit a full application within an institutional category for each of the eight ranked institutional lists used in this study, when comparing the unblinded and blinded reviews. During the unblinded reviews, the range of the Relative Advantage–Full Application for a full application invitation for an LOI from the '1–10' institutional categories was 1.4–1.7 times higher than the average percentage, and the range for the '11–25' institutional categories was 1.2–1.4 times higher, illustrating a consistent bias in favor of more prestigious institutions. After requiring the submission of blinded LOIs, the Relative Advantage–Full Application for an invitation for a full application from both the '1–10' and '11–25' institutional categories decreased to 1.0–1.3 times the average. Importantly, the change in the Relative Advantage–Full Application of receiving a full application invitation for those from the 'Other' institutional category increased from 0.60 to 0.83 times the average during unblinded reviews to 0.81–0.93 with the blinded reviews. This increase is significant as the potentially transformative ideas from junior faculty at these institutions, which might otherwise have been lost to the institutional prestige bias in the unblinded review, are now being considered at the full application review stage. Finally, we found that the ranges for the Relative Advantage–Full Application of receiving

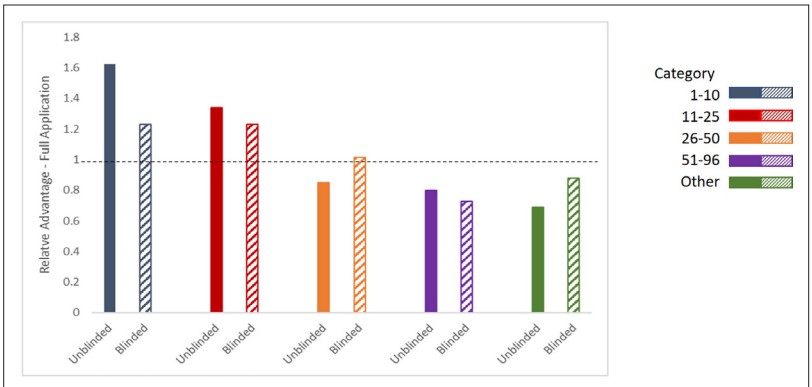

**Figure 2.** Relative Advantage–Full Application with Consensus Institutional Ranking. Ratio of the percentage of LOI applicants in each category in the consensus listing receiving an invitation to submit a Full Application, relative to the percentage of any LOI applicant receiving an invitation to submit a Full Application during unblinded reviews (solid bars, left) and blinded reviews (hatched bars, right; three years of data (2021–2023)).

a full application invitation for applicants in the '26–50' and '51–100' categories remained unchanged between the blinded and unblinded review processes, being 0.83–1.0 times and 0.80–0.95 times lower than the average percentage respectively.

To test for systemic bias, the average value and ranges of the Relative Advantage–Full Applications by institutional category and the results of Chi-squared tests of association between institutional category and full application invitation status within the blinded and unblinded processes for each institutional ranking schema are presented in *Table 2*. The relationship between institutional category and full application invitation status was statistically significant when reviews were unblinded, across all institutional ranking schema. This association represents a medium sized effect (as measured by Cramer's V; range = 0.14–0.23, average = 0.20). After changing to the blinded review process, the relationship between institution category and full application invitation status is not consistently statistically significant (depending on the institutional ranking schema), and the effect size of the association is decreased compared to that under the unblinded review process (as measured by Cramer's V; range 0.08–0.15; average = 0.13). We find that the change to the blinded review process resulted in a consistent decrease in the Chi-squared as well as the Cramer's V statistic for each institutional ranking list, and we conclude that blinded review reduced the impact of institutional prestige bias on full application invitation rates.

In addition to the individual institutional ranking lists used above, we also created a

Consensus Institutional Ranking list by averaging the ranking of the 96 institutions that appeared on at least five of the individual lists. We reasoned that this consensus list might best mirror how our reviewers experience the reading of these individual lists over time and how they consolidate this information into their own heuristic of institutional prestige. We then repeated the calculation of the Relative Advantage–Full Application against the Consensus Institutional Ranking list.

*Figure 2* shows the average Relative Advantage–Full Application afforded to an LOI applicant to receive an invitation to submit a full application within a category in the consensus list, when comparing the unblinded and blinded reviews. During the unblinded reviews, the range of the Relative Advantage–Full Application for a full application invitation for an LOI from the '1–10' and '11–25' categories in the consensus list was 1.6 times and 1.3 times higher than the average percentage respectively, confirming a consistent bias in favor of more prestigious institutions. After requiring the submission of blinded LOIs, the Relative Advantage–Full Application for an invitation for a full application from the '1–10' and '11–25' categories both decreased to 1.2 times the average. Importantly, the change in the Relative Advantage–Full Application of receiving a full application invitation for those from the '26–51' and 'Other' categories again increased from 0.85 times and 0.70 times the average during unblinded reviews, to 1.0 times and 0.88 times the average with the blinded reviews. Finally, consistent with the findings in the individual lists, we found that the Relative Advantage–Full Application of receiving a full application invitation for applicants in the '51–96' category in the consensus list decreased from the blinded and unblinded review processes, from 0.80 times to 0.73 times lower than the average percentage respectively.

The average value and ranges of the Relative Advantage–Full Applications by institutional category in the consensus list, and the results of Chi-squared tests of association between institutional category and full application invitation status within the blinded and unblinded processes, are presented in *Table 3*. As before, the relationship between institutional category and full application invitation status was statistically significant when reviews were unblinded. This association represents a medium sized effect as measured by Cramer's V=0.22. After changing to the blinded review process, the effect size of the association is decreased compared to that under the unblinded review process with

**Table 3.** Relative Advantage–Full Application with Consensus Institutional Ranking.
The average value and ranges by category in the consensus ranking, with Chi-squared association test with Cramer's V statistic of unblinded and blinded LOI reviews for full application invitations.

Ranked List: Consensus

| Category | Unblinded Average (Range) | Blinded Average (Range) | Unblinded - Blinded |
|---|---|---|---|
| 1–10 | 1.6 (1.4–2.0) | 1.2 (1.1–1.4) | 0.4 |
| 11–25 | 1.3 (1.3–1.4) | 1.2 (1.1–1.6) | 0.1 |
| 26–50 | 0.85 (0.73–0.93) | 1.0 (0.90–1.1) | –0.15 |
| 51–100 | 0.80 (0.67–1.0) | 0.73 (0.45–1.0) | 0.07 |
| Other | 0.70 (0.53–0.80) | 0.88 (0.67–1.1) | –0.18 |
| **Analysis** | **Unblinded** | **Blinded** | **Unblinded - Blinded** |
| Chi-squared | 66.87 | 21.9 | 44.97 |
| p (d.f.=4) | 1.04E-13 | 0.00021 | |
| Cramer's V | 0.22 | 0.15 | 0.07 |
| Effect Size | Medium | Medium | |

Cramer's V=0.15. We find that the change to the blinded review process resulted in a consistent decrease in the Chi-squared as well as the Cramer's V statistic. Thus, we conclude that the analysis with the Consensus Institutional Ranking list also shows that the blinded review reduced the impact of institutional prestige bias on full application invitation rates.

We continued the analysis of our review outcomes relative to institutional prestige through to program awards for the study years to determine if the change in institutional representation present in the submitted full applications would extend also to program awards. We calculated the Relative Advantage–Award across all institutional category schema described above: a ratio of the percentage of LOIs submitted in a particular institutional category that received a program award to the percentage of all LOIs that received a program award. As before, if institutional prestige bias did not influence the awardee selection process, then the Relative Advantage–Award would be the same between unblinded and blinded reviews, and we examined the difference in Relative Advantage–Award to determine if there was institutional prestige bias in our selection process. Due to the limited number of program awards that are selected each year, we would expect an average of two awards per category each year if there is no institutional prestige bias. The analysis of program awards with the Consensus Institutional Ranking list is presented here in *Figure 3* and *Table 4*, and the results of the analysis for all institutional categories is presented in *Figure 1—figure*

*supplement 1* and Table S1 in *Supplementary file 1*. In addition, the blinded review data has only three years of awardees, as the fourth year of awardee selection has not been finalized as of manuscript preparation.

*Figure 3* shows the average Relative Advantage–Award afforded to an LOI applicant to receive program award within the categories in the consensus list, when comparing the unblinded and blinded reviews. During the unblinded reviews, the average of the Relative Advantage–Award for an LOI from the '1–10' and '11–25' categories was 2.5 times and 2.0 times higher than the average percentage respectively. With this relative advantage, on average 75% of AMBF's annual program awards were to the top 25 institutions ranked on this list, out of the 287 institutions who applied to the BYI Program during the study years. After requiring the submission of blinded LOIs, the Relative Advantage–Award for a program award from the '1–10' and '11–25' categories decreased to 1.8 times and 1.4 times the average respectively, which represents an average of 45% of the annual program awards. While still an advantage for the top ranked institutions, the decrease in awards to these top institutions open opportunities for the highly rated candidates from other institutions to receive a BYI Program award. The change in the Relative Advantage–Award for those from the '26–50' category increased with the blinded reviews from 0.23 to 0.91 times the average percentage, and the change for the 'Other' category was from 0.42 to 0.83 times the average. Finally, we found that the Relative

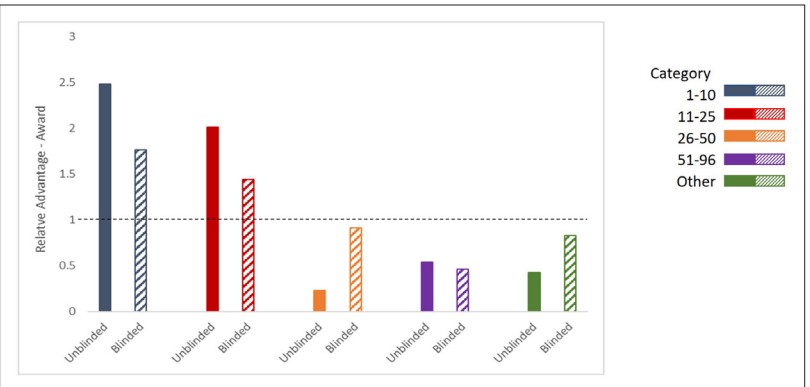

**Figure 3.** Relative Advantage–Awards with Consensus Institutional Ranking. Ratio of the percentage of LOI applicants in each category in the consensus listing receiving a Program Award, compared to the percentage of any LOI applicant receiving a Program Award during unblinded reviews (solid bars, left) and blinded reviews (hatched bars, right; three years of data (2021–2023)).

Advantage–Award for applicants in the '51–96' category decreased moderately from 0.54 to 0.45 times lower than the average percentage during blinded reviews.

Again, the average value and ranges of the Relative Advantage–Award by category in the consensus list, and the results of Chi-squared tests of association between institutional category and program award status, within the blinded and unblinded processes is presented in *Table 4*. The full set of average values and ranges for each institutional category schema is presented in Table S1 in *Supplementary file 1*. The relationship between category in the consensus list and program award status was statistically significant when reviews were unblinded, with a medium sized effect as measured by Cramer's V=0.15. After changing to the blinded review process, the relationship between category and program award status is not statistically significant, and the effect size of the association is decreased compared to that under the unblinded review process. We find that the change to the blinded review process resulted in a decrease in the Chi-squared as well as the Cramer's V statistic for the consensus list, and we conclude that blinded review at the LOI stage also contributed to reducing the impact of institutional prestige bias on program award rates.

### Gender

In addition to institutional prestige bias, we assessed our review process relative to potential gender bias, with assignment of applicants into female or male gender categories as described in the Methods section. *Figure 4* shows the percentage of female applicants that submitted an LOI, received a full application invitation, and received a program award over the years of the study. The percentage of full application invitations to female applicants is consistent with the percentage of LOIs received from female applicants over the years studied, to include unblinded and blinded reviews. The percentage of program awards has a higher variability, but the average of the study years again tracks the percentage of female LOI applicants with no evidence of gender bias in the review process.

Using the same methodology as the analysis of institutional prestige, we calculated the Relative Advantage–Gender as the ratio of the percentage of each gender to receive a full application invitation or program award relative to the percentage of any applicant to receive a full application invitation or program award. *Table 5* presents the average value and ranges of the Relative Advantage–Gender for the two gender categories and the results of Chi-squared tests of association between gender category and full application invitation or program award status within the blinded and unblinded processes. The relationship between gender category and full application invitation or program award status is not statistically significant, and there is no effect of gender bias from the review process as seen by the Cramer's V of 0.00–0.03 across all conditions studied. We therefore conclude that our review process and subsequent program awards did not demonstrate gender bias, either before or after blinding the LOI reviews.

### Discussion

Overall, this work illustrates one of the major challenges AMBF and other funders face when supporting young scientists. We seek to find the exceptional individuals conducting transformative science at all institutions, not just those that are at institutions perceived by reviewers to have higher prestige. However, in the drive to support the most exciting and innovative researchers and ideas, it is apparent that the qualitative metric of institutional affiliation (whether explicit or implicit) often used in applicant assessment may not be a good proxy for scientific innovation. The bias we found towards prestigious institutions in the unblinded reviews reveals that faculty at institutions with a lesser reputation are not afforded the same allowances from reviewers as those applicants at institutions with a greater reputation.

We observed that the institutional prestige in our review process was reduced, but not

**Table 4.** Relative Advantage–Award with Consensus Institutional Ranking.
The average value and ranges for consensus categories, with Chi-squared association test and Cramer's V statistic of unblinded and blinded LOI reviews through program awards. Analysis of blinded reviews relied on three years of data (2021–2023).

Ranked List: Consensus

| Category | Unblinded Average (Range) | Blinded* Average (Range) | Unblinded - Blinded |
|---|---|---|---|
| 1–10 | 2.5 (1.6–3.4) | 1.8 (0.63–2.6) | 0.7 |
| 11–25 | 2.0 (0.61–3.3) | 1.4 (1.1–2.6) | 0.6 |
| 26–50 | 0.23 (0.0–0.52) | 0.91 (0.70–1.1) | –0.68 |
| 51–100 | 0.54 (0.0–1.0) | 0.46 (0.0–0.83) | 0.08 |
| Other | 0.42 (0.33–0.68) | 0.83 (0.42–0.90) | –0.41 |
| Analysis | Unblinded | Blinded* | Unblinded - Blinded |
| Chi-squared | 30.62 | 5.18 | 25.44 |
| p (d.f.=4) | 3.66E-06 | 0.269 | |
| Cramer's V | 0.15 | 0.09 | 0.06 |
| Effect Size | Medium | Small | |

eliminated, by blinding the LOI applications, indicating that there may be other bona fide measurable differences between institutional categories. The origin of this difference may be a combination of the quality of the candidates themselves, the physical research infrastructure of their universities, and the resources and support available to junior faculty at well-resourced institutions, including in submitting applications to funding opportunities. There also may have been additional factors that influenced our results between the application cycles. For example, we did receive fewer overall applications in blinded years of our study due to COVID-19 research disruptions and hiring freezes across institutions. However, LOIs were received in each institutional category consistently across the studied years indicating that our initial populations were composed of LOIs with similar institutional diversity, as shown in *Table 6*. Additionally, we informed our reviewers at the beginning of the review process that the purpose in blinding the LOIs was in part to study if there was institutional prestige bias in our reviews. This awareness may have impacted the scoring of proposals if the reviewers were more willing to advance LOIs or full applications that had weaknesses attributed to lack of mentorship or access to resources, which might indicate that the applicant was from an institution that did not have the same level of research support infrastructure.

There is no universally accepted ranking of institutions and therefore there are many possible methods that can be used to define and measure the conceptual quality of institutional prestige. Every reviewer brings to the review process their own unique perception of institutional prestige based on the qualities that they value most which may also influence any implicit or explicit bias. For this study, we chose several lists that used different measurable variables in their institutional rankings, including federal research funding received, research publications and citations by faculty, prestigious awards received by faculty, and student outcomes, among other variables. These variables were likely important to our reviewers and are relevant to our mission of supporting basic science, however they may or may not have covered all possible variables or may have been too restrictive. For example, the Leiden ranking list had the lowest correlation with our review outcomes, possibly because the filter settings we used for that list included publications within biomedical and related journals, which may not have accurately reflected our total applicant and reviewer pool that come from disciplines across chemistry, biology, and life sciences. Also, the ranked list based on historical AMBF funding was not provided to any reviewers or previously published but was included because of our own internal reviewer recruitment practices.

Anecdotally, reviewers who participated in the BYI LOI reviews reported that the blinded materials were easier to read, that there was a significant reduction in their workload to review the same number of LOIs, and the review meetings were shorter as it was easier to focus solely on the merits of the technical proposal during

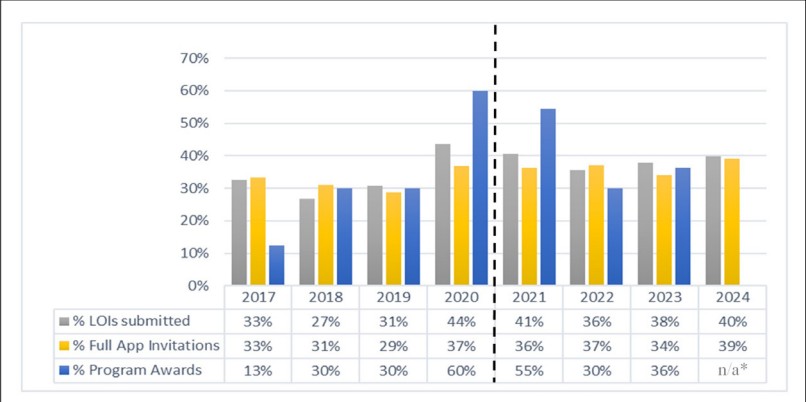

**Figure 4.** Outcomes for female applications. Percentage of female LOI applicants to receive a Full Application invitation and Program Award by year. Between 2017 and 2020 the reviews of initial LOIs were not blinded; from 2021 onwards the reviews of initial LOIs were blinded; Program Awards for 2024 had not been finalized as of manuscript preparation.

The online version of this article includes the following source data for figure 4:

**Source data 1.** BYI Letters of Intent (LOIs), Full Application Invitations, and Program Awards by gender category.

the discussion. With these observed benefits in reducing institutional prestige bias and the positive comments from our reviewers, AMBF plans to continue to use the blinded review for the BYI LOI application step, and we have also adopted these same review methods to our Postdoctoral Fellowship program application process. In sharing these results, AMBF hopes this information will be informative for others moving forward with evaluation of their application review processes with the goal of instituting more equitable practices (*Franko et al., 2022*), especially

for those organizations with missions and funding programs similar to our own.

### Future directions

AMBF will continue to monitor our application metrics as we strive to continuously improve our own internal methods and processes for a fair evaluation process for any applicant to our programs. AMBF is also increasing our outreach and materials available to assist applicants and encouraging applicants interested in our programs from institutions that may not have the same level of internal resources to consider applying. Finally, AMBF will continue to diversify our reviewer cohort to ensure multiple perspectives are included at all levels of the review process.

## Materials and methods

### Experimental design

The AMBF BYI Program application process, shown in *Figure 5*, starts with an open call for LOIs from junior faculty who are within the first four years of a tenure-track appointment at US institutions, followed by a limited number of invited full applications selected from the submitted LOIs. The Foundation broadly distributes an announcement to research institutions annually when the program opens, and all US institutions that have tenure track, or equivalent, positions are encouraged to apply. The LOI review step was selected as the focus of this

**Table 5.** Relative Advantage–Gender.
The average value and ranges for gender categories, with Chi-squared association test and Cramer's V statistic of unblinded and blinded LOI reviews in full application invitations and program awards. Analysis of blinded reviews relied on three years of data (2021–2023).

| Full Application Invitations | | | | Program Awards | | | |
|---|---|---|---|---|---|---|---|
| Category | Unblinded Average (Range) | Blinded Average (Range) | Unblinded - Blinded | Category | Unblinded Average (Range) | Blinded* Average (Range) | Unblinded - Blinded |
| Male | 1.0 (0.94–1.1) | 1.0 (1.0–1.1) | 0 | Male | 0.99 (0.71–1.3) | 0.96 (0.77–1.1) | 0.03 |
| Female | 0.99 (0.84–1.2) | 0.93 (0.90–0.98) | 0.06 | Female | 0.96 (0.38–1.4) | 1.0 (0.84–1.3) | –0.04 |
| Analysis | Unblinded | Blinded | Unblinded - Blinded | Analysis | Unblinded | Blinded* | Unblinded - Blinded |
| Chi-squared | 0.06 | 0.88 | –0.82 | Chi-squared | 0.0 | 0.01 | –0.01 |
| p (d.f.=1) | 0.799 | 0.347 | | p (d.f.=1) | 1.0 | 0.916 | |
| Cramer's V | 0.01 | 0.03 | –0.02 | Cramer's V | 0.00 | 0.00 | 0.00 |
| Effect Size | No effect | No effect | | Effect Size | No effect | No effect | |

**Table 6.** Percentage of LOIs Received per Institutional Category for the eight institutional ranking and the consensus ranking.

| Institutional Category | NSF NCSES 2018 | NSF NCSES 2020 | Times Higher 2018 | Times Higher 2023 | Shanghai Ranking 2018 | Shanghai Ranking 2023 | Leiden 2018–2021 | AMBF 1990–2018 | Consensus |
|---|---|---|---|---|---|---|---|---|---|
| 1–10 | 13% | 13% | 14% | 14% | 13% | 15% | 12% | 15% | 15% |
| 11–25 | 21% | 21% | 19% | 19% | 19% | 18% | 18% | 19% | 18% |
| 26–50 | 20% | 19% | 24% | 26% | 30% | 24% | 18% | 18% | 24% |
| 51–100 | 20% | 20% | 19% | 18% | 19% | 23% | 20% | 17% | 20% |
| Other | 25% | 27% | 23% | 23% | 18% | 20% | 32% | 30% | 23% |

analysis as it is the first gate in the application process, and it is the step with the most applications and therefore the highest workload for our reviewers. Past studies have shown that implicit bias is strongest when reviewer workload is high (*Kahneman, 2011*). The unblinded LOI included a three-page technical proposal, the applicant's bio sketch, and a chart of external funding either pending or received. During unblinded reviews, all three documents were provided to reviewers. When we transitioned to blinded reviews, the LOI was extended to a four-page blinded technical proposal which was shared with reviewers, and the applicant's bio sketch, self-reported gender and ethnicity information, and external funding charts were collected by the Foundation and used internally only.

For the blinding process, we provided instructions for applicants to blind their own technical proposals prior to submission by not including applicant name, gender, gender-identifying pronouns, or institutional information in their technical proposal, along with specific formatting to follow for referencing publications (*AMBF, 2023*). Submitted LOIs were reviewed by Foundation staff for eligibility and compliance with the blinding process prior to assigning LOIs to review panels through our online portal. This method was used to reduce the administrative burden on AMBF staff members; staff did not edit or modify any submitted LOIs to comply with the blinding requirements. We found that compliance with the blinded technical proposal preparation was very high, and those applicants who blatantly did not follow the blinding rules were generally ineligible to apply based on other Foundation criteria. All eligible LOIs were assigned a random 4-digit number to be used as the proposal identifier for the reviewers throughout the LOI review process and discussions.

In compliance with the Foundation's standard practice, reviewers were recruited from the population of tenured researchers at US institutions, also including past BYI award recipients. Panels composed of three reviewers were assigned sets of eligible LOIs to review, while ensuring that no reviewer was assigned an applicant from their same institution. For all program years in this study, written reviewer scores and comments were collected from each reviewer independently and then LOI review panel calls were held via teleconference to discuss the merits and concerns of the LOIs and select those to advance to full application. Each panel was instructed on the number of LOIs they could recommend advancing to the Full Application stage and each panel completed their selections independently of the other panels. The outcome of the LOI review each year was to select about 100 total applicants who were then invited to submit full applications.

During the blinded review process, reviewers were asked to assess the merits of the technical proposal as well as to confirm that the blinding rules were followed by the applicant. If any reviewer identified an applicant as not complying with the blinding rules, that LOI was discussed at the start of the review panel meeting and could be disqualified from consideration with a majority vote of the review panel. Notably, an application was not disqualified if a reviewer believed they could infer the identity of the author as long as the blinding rules were followed. Only four LOIs were eliminated from consideration by review panel vote in the years of blinded review.

The process of full application submission and review remained similar during all years of the study. The selected LOI applicants were invited to submit a full application which consisted of an updated six-page research proposal, proposed budget and timeline, bio sketch, letters of recommendation from advisors and colleagues, and institutional support forms. The full applications were reviewed by four panels of three reviewers each, composed of a subset of the reviewers who had assisted in the LOI review stage. Two of

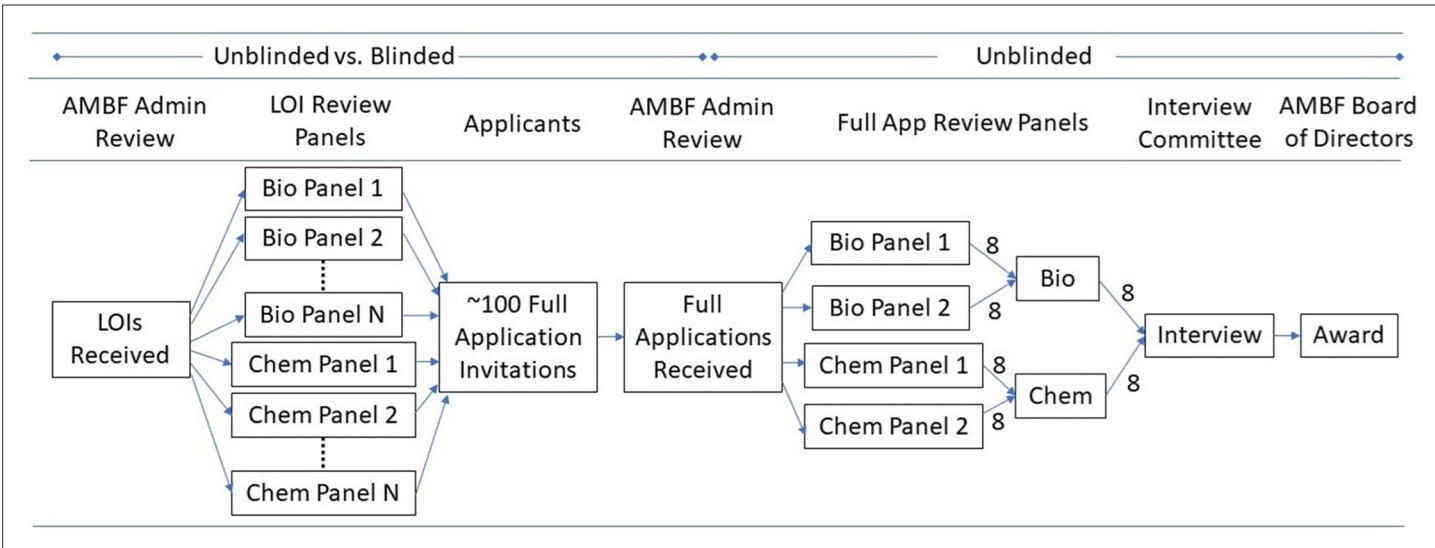

**Figure 5.** Schematic of the BYI Application Review Process.

the panels reviewed full applications in biology related fields and the other two panels reviewed full applications in chemistry related fields. Written reviewer scores and comments were collected from each reviewer independently and then the biology and chemistry panels convened for a day and a half to discuss each proposal. On the first day, each of the four panels selected their top eight full applications, for a total of 16 advancing biology full applications and 16 advancing chemistry full applications. On the second day, these full applications were discussed within combined biology and chemistry panels to select the top 16 candidates to invite to an interview. The interviews with the candidates were conducted with members of the BYI Program Executive Committee, in-person through 2019 and virtually in subsequent years. The Interview Committee provided recommendations for program awards to the AMBF Board of Directors for final approval.

### Institutional categories

To study if there was a change in the institutional prestige of selected LOIs before and after blinding the review process, we developed a method to assign each LOI into an institutional category related to the institutional prestige of the applicant's affiliated institution, including associated medical schools and research institutes. There are several rankings and lists of universities and research institutions that use metrics that could be associated with prestige within the scientific research community. We examined the impact of our review process against seven different ranked lists, over multiple years, as well as a ranking of

institutions that have historically received funding from AMBF. We reasoned that these lists, which encompass research funding, publications, faulty awards, and teaching, would include the universities and colleges with chemistry, life sciences, engineering, and interdisciplinary research programs that had recently applied for and received science funding through similar peer review process as ours and are therefore institutions likely identified as having prestigious reputations by our reviewers.

Table S2 in *Supplementary file 2* contains, in rank order, the top 100 institutions for each of the eight institutional lists used in this study:

- Lists based on data released by the National Center for Science and Engineering Statistics (NCSES), which is part of the National Science Foundation (NSF). The data we used was for federal obligations for science and engineering funding to universities and colleges in 2018 (*NSF, 2018a*) and in 2020 (*NSF, 2020*). Data comes from an annual survey in which federal funding agencies are asked to report their obligations for science and engineering funding (*NSF, 2018b*).
- The Shanghai Academic Ranking of World Universities, which annually ranks universities by several academic or research performance indicators, including alumni and staff winning Nobel Prizes and Fields Medals, highly cited researchers, papers published in *Nature* and *Science*, papers indexed in major citation indices, and the per capita academic performance of an institution (*Shanghai Ranking, 2023*). The

list used in this study was filtered for universities in the US, and for years 2018 and US, and 2023. The top 100 ranked universities (including ties) were included.

- The Times Higher Education World University Rankings are based on 13 calibrated performance indicators that measure an institution's performance across four areas: teaching, research, knowledge transfer and international outlook (***Times Higher Education, 2023***). The list used in this study was filtered for Research Universities in the United States, and for years 2018 and 2023. The top 100 ranked universities were included.
- The Leiden Ranking, published by Centre for Science and Technology Studies at Leiden University, is based on bibliographic data on scientific publications, in particular on articles published in scientific journals using Web of Science as the primary data source (***CWTS, 2023***). The list used in the study was filtered for time period 2018–2021, discipline of 'Biomedical and health sciences', United States, and sorted by scientific impact.
- A ranked list of funding received by Institution from AMBF in the years 1990–2018, including funding from the BYI Program, the Beckman Scholars Program, and the Arnold O Beckman Postdoctoral Fellows Program. We often recruit reviewers from our past awardees, which may have introduced an implicit bias to these institutions from the structure of our reviewer pool.

For the NCSES, Shanghai and Times Higher lists, the ranks for the universities did not change appreciably between the years examined. Among the lists, there were some universities with consistent ranks, such as Johns Hopkins University with ranks between 1 and 15, and University of Virginia with ranks between 46 and 62. However, some universities had much larger discrepancies among the lists, such as California Institute of Technology with ranks of 2, 7, 24, 53, 66 and not appearing in the top 100 on one list. As an additional ranked list to use in the study, we created a 'consensus' list by averaging the ranks for the 96 institutions that appeared on a majority (at least five) of the selected lists (see Table S2 in ***Supplementary file 2***). We divided each list into five institutional categories, with the top category including just 10% of the ranked institutions, as the top institutions disproportionately secure most of the research funding (***Lauer and Roychowdhury, 2021***; ***NSF, 2018a***; ***NSF, 2020***):

- '1–10': The first ten institutions in the ranked list.

- '11–25': The next 15 institutions in the ranked list.
- '26–50': The next 25 institutions in the ranked list.
- '51–100': The remaining 50 institutions in the ranked list; or 51–96 for the consensus list.
- 'Other': The institutions that applied to the AMBF BYI Program during the study years that were not included in the categories above.

As we often receive more LOI applications from the institutions ranked highly on these lists, the five categories also divides the LOIs received into each of these categories. ***Table 6*** presents the percentage of LOIs received in each institutional category, averaged across all study years, by ranking list.

After defining the institutional categories, we assigned each LOI into an institutional category based on the applicant's affiliation and then we analyzed the number of LOIs received in each institutional category, the number of full application invitations in each institutional category, and the number of program awards made by the Foundation in each institutional category. The full list of LOIs, Full Application Invitations, and Program Awards by year is included in Table S3 in ***Supplementary file 3***, sorted alphabetically by the 287 institutions that submitted LOIs during the study years. If there was no bias towards institutional prestige in our reviews, then we would expect that the percentage of LOIs that advance to a full application invitation and to a program award within each institutional category relative to the total percentage of LOIs that advance to a full application invitation and to a program award would be the same before and after the blinded reviews.

### Gender
To study if there was a gender bias based on applicant characteristics in our review process before and after blinding the reviews, we assigned each LOI to a category of female or male. For the unblinded years, applicants were not asked to self-identify their gender in the application materials and the data presented are from AMBF research of applicant name and affiliation to associate each LOI into a category of female or male. For the blinded application years, applicants did self-identify their gender during LOI submission, and the 'female' category includes applicants that identify as female, and the 'male' category includes applicants that identify as male, transmale (self-reported by one LOI applicant), and

non-binary (self-reported by five LOI applicants, all of whom have traditionally male first names). If there was no bias towards gender in our reviews, then we would expect that the percent of LOIs that advance to a full application invitation and to a program award within each gender category would equal the total percentage of LOIs that advance to a full application invitation and to a program award.

### Statistical analysis

A Chi-squared test for independence (*McHugh, 2013*) was used to examine the association between institutional category and invitation status (invited to submit a full application, or not invited to submit a full application) and award status (awarded, or not awarded), as well as gender and invitation status (invited to submit a full application, or not invited to submit a full application) and award status (awarded, or not awarded). Observed frequencies were compared to the expected frequencies, calculated based on the average invitation and award rate for all LOIs. Results of statistical analyses were considered statistically significant at $P<0.05$. Cramer's V statistic is also provided as a measurement of the effect size for the Chi-squared tests.

### Acknowledgements

We thank everyone who has reviewed applications for the BYI. We also thank the Board of Directors of the AMBF for supporting the trial reported here, and our Scientific Advisory Council and the BYI Executive Committee for their help and support. We also thank Amanda Casey and Dr Kim Orth (University of Texas Southwestern), and Dr Michael May (Dana Point Analytics) for assistance with data analysis and for visualization recommendations.

**Anne E Hultgren** is at the Arnold and Mabel Beckman Foundation, Irvine, United States
ahultgren@beckman-foundation.org
ⓘ https://orcid.org/0009-0001-8266-1050
**Nicole MF Patras** is at the Arnold and Mabel Beckman Foundation, Irvine, United States
ⓘ https://orcid.org/0009-0008-6229-0038
**Jenna Hicks** is at the Health Research Alliance, Research Park, United States
ⓘ https://orcid.org/0000-0002-4486-9926

*Author contributions:* Anne E Hultgren, Conceptualization, Data curation, Methodology, Writing – original draft; Nicole MF Patras, Investigation, Methodology; Jenna Hicks, Formal analysis, Writing – review and editing

*Competing interests:* The authors declare that no competing interests exist.

### Funding

No external funding was received for this work.

**Decision letter and Author response**
Decision letter https://doi.org/10.7554/eLife.92339.sa1
Author response https://doi.org/10.7554/eLife.92339.sa2

# Additional files

## Supplementary files

• Supplementary file 1. Relative Advantage–Award. The average value and ranges for institutional categories, with Chi-squared association test and Cramer's V statistic for unblinded LOI reviews (2017–2020) and blinded LOI reviews (2021–2023) through program awards.

• Supplementary file 2. Ranked lists of institutions. The top 100 Institutions in the eight ranked lists used in the study, plus the consensus ranking list.

• Supplementary file 3. BYI LOIs, Full Application invitations, and Program Awards by Institution. All LOIs received, Full Application invitations, and Program Awards from 2017 to 2020 (unblinded) and 2021–2024 (blinded), sorted alphabetically by institution name.

• MDAR checklist

• Source code 1. R-notebook for Chi-squared analysis.

## Data availability

All data generated or analysed during this study are included in the manuscript and associated source data files.

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
