## [Decision Letter]

**Decision letter after peer review:**

Thank you for submitting your article "Blinding Reduces Institutional Prestige Bias During Initial Review of Applications for a Young Investigator Award" to *eLife* for consideration as a Feature Article. Your article has been reviewed by two peer reviewers, and the evaluation has been overseen by Peter Rodgers of the *eLife* Features Team. Both reviewers have opted to remain anonymous.

The reviewers and editors have discussed the reviews and we have drafted this decision letter to help you prepare a revised submission. Addressing the comments listed below will require substantial work, so you may prefer to submit the manuscript to a different journal: if you decide to do this, please let me know.

Summary:

In this manuscript the authors report that blinding the identity of applicants during the first round of review for applications for a Beckman Young Investigator Award led to a reduction in prestige bias. This topic is of central importance for fair scientific funding allocation. The manuscript is clearly written, and some of the key limitations are openly discussed. However, there are a number of concerns that need to be addressed, notably concerning the proxy for prestige used by the authors. Moreover, the authors' dataset could also be used to explore possible gender bias in the review for these awards.

Essential revisions

1. The manuscript, as it stands, does not bring much more to the topic than what has been published in previous papers, such as Solans-Domenech et al., 2017, which analyzed a larger sample (and is not cited by the authors – see point 4 below). In order to improve the manuscript I would advise the authors to expand the scope as follows:

1a. The option and justification given by the authors of not studying the effect of gender is not convincing. Although applicants were not asked about their gender before 2021, it could still be possible to perform an analysis based on a proxy where first names would be associated to gender.

1b. The following point is optional: It would be interesting to check the final outcome of the evaluation, at the second step, when the proposal is not blinded anymore, and to assess if the final set of successful candidates (in terms of affiliations) differ from those before the blinding step was introduced.

2. The prestige ranking is based on total funding size, which is certainly correlated with the prestige of the institutions. However, this measure is also highly correlated with the size of the institutions, which might not accurately reflect the prestige: for example, Harvard, Yale and MIT are in the second tier; Chicago, Berkeley and Caltech are in the third tier; and there are also some very prestigious institutions (such as Rockefeller University, Fred Hutchinson Cancer Research Center, and the Mayo Clinic) in the bottom tier.

This point could be addressed by using a modified prestige ranking obtained by normalizing the funding size with respect to the size of the institutions (e.g., the number of researchers) might be more appropriate.

Another approach would be to use other university rankings (eg Shanghai, THE, Leiden, etc).

Our suggestion is that several different proxies for prestige should be used in the study (eg two or more of NSF, NSF normalized, Shanghai, THE, Leiden, and maybe even the proxy based on previous performance when applying to Beckman), and that the results of all the proxies studied should be reported.

3. There are a number of points regarding the statistical analysis that need to be addressed

3a. Regarding the NSF ranking: the 2018 data has been used, but as the study covers 8 years, the authors should at least have tested if rankings from different years would give different results.

3b. The division of prestige categories into four groups based on equal partition of total national funding, while reasonable, still seems a bit arbitrary. It might be of interest to test the robustness of the results with respect to the number of categories chosen.

3c. The sample size in the treatment group (data from 2021-2023) is substantially smaller than the sample size in the control group (data from 2016-2020). Therefore, the fact that the p-value becomes insignificant might simply be because of reduced sample size instead of reduced prestige bias. Down-sampling the institutions for 2016-2020 might make the comparison more fair. Or alternatively, a difference-in-difference design might be able to show the effect more clearly.

4. The discussion of the existing literature on bias in research grant evaluation needs to be improved. I recommend that the authors cite and discuss the following articles:

Forscher, P.S., Cox, W.T.L., Brauer, M. et al. (2019) Little race or gender bias in an experiment of initial review of NIH R01 grant proposals. Nat Hum Behav 3:257-264.

Marsh et al. (2008) Improving the peer-review process for grant applications: Reliability, validity, bias, and generalizability. American Psychologist, 63:160-168.

Solans-Domenech et al. (2017) Blinding applicants in a first-stage peer-review process of biomedical research grants: An observational study. Res. Eval. 26:181-189. DOI:10.1093/reseval/rvx021

I also recommend that the authors cite and discuss the following article about prestige bias in the peer review of papers submitted to a conference:

Tomkins et al. (2017) Reviewer bias in single-versus double-blind peer review. PNAS 114:12708-12713.

---

## [Author Response]

Essential revisions1. The manuscript, as it stands, does not bring much more to the topic than what has been published in previous papers, such as Solans-Domenech et al., 2017, which analyzed a larger sample (and is not cited by the authors – see point 4 below). In order to improve the manuscript I would advise the authors to expand the scope as follows:1a. The option and justification given by the authors of not studying the effect of gender is not convincing. Although applicants were not asked about their gender before 2021, it could still be possible to perform an analysis based on a proxy where first names would be associated to gender.

The analysis on gender distribution in the LOI, Full Application invitations, and Program Awards before and after blinding is included in the revision. In collecting this data, we found that there was no evidence of gender bias in the Full Application invitations or Program Awards either before or after blinding.

1b. The following point is optional: It would be interesting to check the final outcome of the evaluation, at the second step, when the proposal is not blinded anymore, and to assess if the final set of successful candidates (in terms of affiliations) differ from those before the blinding step was introduced.

We extended our analysis to include Program Awards based on institutional affiliation and included the results in the discussion.

2. The prestige ranking is based on total funding size, which is certainly correlated with the prestige of the institutions. However, this measure is also highly correlated with the size of the institutions, which might not accurately reflect the prestige: for example, Harvard, Yale and MIT are in the second tier; Chicago, Berkeley and Caltech are in the third tier; and there are also some very prestigious institutions (such as Rockefeller University, Fred Hutchinson Cancer Research Center, and the Mayo Clinic) in the bottom tier.This point could be addressed by using a modified prestige ranking obtained by normalizing the funding size with respect to the size of the institutions (e.g., the number of researchers) might be more appropriate.Another approach would be to use other university rankings (eg Shanghai, THE, Leiden, etc).Our suggestion is that several different proxies for prestige should be used in the study (eg two or more of NSF, NSF normalized, Shanghai, THE, Leiden, and maybe even the proxy based on previous performance when applying to Beckman), and that the results of all the proxies studied should be reported.

We implemented the suggestion presented here from the reviewers and expanded our analysis to include eight different lists of institutional ranks, as well as then developed a “Consensus List” based on reviewing the trends and differences between these lists. The analysis results using all nine of these lists are included in the revised manuscript.

With regards to normalizing the NCSES Federal funding data by institution size, we did consult with the program team at NSF NCSES to ask if they also collect data that could help with this analysis. While one of their surveys they send to institutions annually (the HERD survey) does include a question about the number of faculty engaged in scientific research, they commented that the compliance with response to this question is lower than usual, and not all of the institutions on the top 100 list would be included in this HERD survey (private communication). With the inclusion of the other university rankings, we did not pursue this question about normalization any further.

3. There are a number of points regarding the statistical analysis that need to be addressed3a. Regarding the NSF ranking: the 2018 data has been used, but as the study covers 8 years, the authors should at least have tested if rankings from different years would give different results.

We included additional years of rankings in our study and discussion.

3b. The division of prestige categories into four groups based on equal partition of total national funding, while reasonable, still seems a bit arbitrary. It might be of interest to test the robustness of the results with respect to the number of categories chosen.

We respectfully disagree that this division was based only on the concentration of funding received at the top institutions, although that is a real effect that is reported in the references cited. The number of categories was also based on the division of our LOIs into these categories (see Table 6 in the Methods section). We also added Supplemental Table 3 with the full data set of number of LOIs, Full Application invitations and Program Awards by institution to help the reader see how the applications are distributed among the institutions each year.

3c. The sample size in the treatment group (data from 2021-2023) is substantially smaller than the sample size in the control group (data from 2016-2020). Therefore, the fact that the p-value becomes insignificant might simply be because of reduced sample size instead of reduced prestige bias. Down-sampling the institutions for 2016-2020 might make the comparison more fair. Or alternatively, a difference-in-difference design might be able to show the effect more clearly.

For this revision, we were able to include an additional year of blinded review analysis, and we balanced the study with 4 years of unblinded and 4 years of blinded data. We did still receive a smaller number of LOIs during some of the blinded study years (which we attribute to COVID disruptions). We considered this difference in sample size in developing the “Relative Advantage” metric that we used for our analysis. If you down-sample the LOIs received in the unblinded years across the Institutional Categories, the “Relative Advantage” ratio remains the same.

4. The discussion of the existing literature on bias in research grant evaluation needs to be improved. I recommend that the authors cite and discuss the following articles:Forscher, P.S., Cox, W.T.L., Brauer, M. et al. (2019) Little race or gender bias in an experiment of initial review of NIH R01 grant proposals. Nat Hum Behav 3:257-264.Marsh et al. (2008) Improving the peer-review process for grant applications: Reliability, validity, bias, and generalizability. American Psychologist, 63:160-168.Solans-Domenech et al. (2017) Blinding applicants in a first-stage peer-review process of biomedical research grants: An observational study. Res. Eval. 26:181-189. DOI:10.1093/reseval/rvx021I also recommend that the authors cite and discuss the following article about prestige bias in the peer review of papers submitted to a conference:Tomkins et al. (2017) Reviewer bias in single-versus double-blind peer review. PNAS 114:12708-12713.

Thank you for the additional references, which have been consulted and included in the revised manuscript.